# The Computational Limits of State-Space Models and Mamba via the Lens of Circuit Complexity

Yifang Chen[1], Xiaoyu Li[2], Yingyu Liang[3,4], Zhenmei Shi[3], Zhao Song[5]

[1]The University of Chicago, [2]University of New South Wales,
[3]University of Wisconsin-Madison, [4]The University of Hong Kong,
[5]The Simons Institute for the Theory of Computing at UC Berkeley

yifangc@uchicago.edu, xiaoyu.li2@student.unsw.edu.au, yingyul@hku.hk,
yliang@cs.wisc.edu, zhmeishi@cs.wisc.edu, magic.linuxkde@gmail.com

In this paper, we analyze the computational limitations of Mamba and State-space Models (SSMs) by using the circuit complexity framework. Despite Mamba's stateful design and recent attention as a strong candidate to outperform Transformers, we have demonstrated that both Mamba and SSMs with $\mathrm{poly}(n)$-precision and constant-depth layers reside within the DLOGTIME-uniform $\mathsf{TC}^0$ complexity class. This result indicates Mamba has the same computational capabilities as Transformer theoretically, and it cannot solve problems like arithmetic formula problems, boolean formula value problems, and permutation composition problems if $\mathsf{TC}^0 \neq \mathsf{NC}^1$. Therefore, it challenges the assumption Mamba is more computationally expressive than Transformers. Our contributions include rigorous proofs showing that Selective SSM and Mamba architectures can be simulated by DLOGTIME-uniform $\mathsf{TC}^0$ circuits, and they cannot solve problems outside $\mathsf{TC}^0$.

## 1. Introduction

Sequential neural networks like RNNs, including their variants such as LSTMs and GRUs [1, 2], have good performance in capturing temporal dependencies and processing input step-by-step [3]. These advantages make them effective in tasks including time-series prediction [4] and speech recognition [5]. Traditional RNNs [6] and their enhanced variance, LSTMs perform well in testing because of their sequential nature, but their training times tend to be slow and suffer from vanishing or exploding gradient issues, which limit their capabilities to capture long-term dependencies [7]. Transformers [8], equipped with a self-attention mechanism, provides an efficient solution to the slow training problem by enabling parallelized computations. Large Language Models (LLMs) based on the Transformer architecture, such as GPT-4 [9], GPT-4o [10], OpenAI's o1 [11], Llama 3.1 [12], Claude [13], and Gemini [14], have become ubiquitous nowadays, and their integrations into modern technology reshaped our expectations of the limits of their capabilities. Transformers are capable of training efficiently on large datasets, but their quadratic memory and time complexity with respect to sequence length make them expensive in resources, both in terms of memory and processing power, during training and inference. Specifically, self-attention mechanisms grows $O(n^2)$ in terms of computational complexity [15].

State-space models (SSMs) recently received significant attention as a potential alternative to Transformer-based architecture on inherently sequential tasks [16]. Mamba [17, 18], built on SSMs, combines the benefits from both RNNs and Transformers architectures. Mamba incorporates the efficient inference and state-tracking capabilities of RNNs and leverages the scalability and parallelizable computations of Transformers. Equipped with long-term memory embedding, Mamba balances the trade-off between training efficiency and inference performance [17].

As these architectures continue to express the state of modern AI, it is crucial to explore what types of problems they can solve and their limitations. Recent studies using the circuit complexity framework explain the computational capabilities of Mamba. [19] demonstrates that a threshold circuit with constant depth and $c \log n$-precision can simulate depth $d$ SSM and Mamba. Moreover, an L-uniform

threshold circuit of constant depth can simulate such SSM and Mamba models. Another work [20] shows Transformers are in DLOGTIME-uniform $\mathsf{TC}^0$ with $\mathrm{poly}\, n$-precision, and they present a new set of metrics to evaluate the circuit complexity of LLMs with $\mathrm{poly}\, n$-precision. Understanding Mamba's computational limits with high precision is crucial because we need to know what problems it can theoretically solve and to compare Mamba with Transformers and other architectures. Without such understanding, assumptions about Mamba's potential to surpass Transformers in terms of sequential reasoning or state tracking remain questionable.

Table 1: Circuit Complexity of SSM/Mamba. Previous work [19] claims a L-uniform threshold circuit of constant depth can simulate SSM/Mamba with $c \log n$-precision, whereas Theorem 4.4 and 4.5 improve the precision and uniformity by proving a DLOGTIME-uniform $\mathsf{TC}^0$ threshold circuit of constant depth can simulate SSM/Mamba with $\mathrm{poly}(n)$-precision.

| Reference | Precision | Circuit Complexity |
|---|---|---|
| Theorem 4.4 of [19] | $c \log(n)$-precision | L-uniform $\mathsf{TC}^0$ |
| Our Theorems 4.4 and 4.5 | $\mathrm{poly}(n)$-precision | DLOGTIME-uniform $\mathsf{TC}^0$ |

However, from Table 1, prior work [19] primarily focused on low-precision implementations or alternative uniformity conditions, leaving a gap in understanding Mamba's expressiveness with $\mathrm{poly}(n)$-precision under DLOGTIME-uniformity. This gap is significant because proving Mamba in $\mathsf{TC}^0$ with $\mathrm{poly}(n)$-precision reflects real-world scenarios, where higher precision is often necessary. Moreover, DLOGTIME-uniformity is widely considered as a more realistic condition in practice. Unlike L-uniform circuits, which may allow unrealistically complex preprocessing, DLOGTIME-uniform circuits require the structure of the circuit to be computable by highly efficient machines, so DLOGTIME-uniformity reflects practical constraints on constructing and applying the circuits. Therefore, it is natural to raise the question: *Can Mamba, implemented with $\mathrm{poly}(n)$-precision, be proved to reside within* DLOGTIME-*uniform* $\mathsf{TC}^0$?

In this paper, we break down the fantasized superiority in Mamba by demonstrating that it falls within the same circuit complexity class DLOGTIME-uniform $\mathsf{TC}^0$ with $\mathrm{poly}\, n$-precision. This result shows SSM and Mamba have the same computational capabilities as Transformers have [20], indicating that SSM and Mamba, despite their stateful design, cannot solve problems outside $\mathsf{TC}^0$, such as arithmetic formula problem, boolean formula value problem, and permutation composition problems if $\mathsf{TC}^0 \neq \mathsf{NC}^1$.

Beyond [19] and [20], our contributions are summarized as follows: If $\mathsf{TC}^0 \neq \mathsf{NC}^1$, assume we have the $\mathrm{poly}(n)$-bits precision float point number, constant-depth layers, and $O(n)$ size hidden dimension, then we have

- A DLOGTIME-uniform $\mathsf{TC}^0$ circuit family can simulate Selective SSM (Theorem 4.4).

- A DLOGTIME-uniform $\mathsf{TC}^0$ circuit family (Theorem 4.5) can simulate Mamba.

- Selective SSM and Mamba are not capable of resolving the arithmetic formula problems, Boolean formula value problems, and permutation composition problems (Theorem 5.1).

Knowing the true computational capabilities of SSM and Mamba in DLOGTIME-uniform $\mathsf{TC}^0$ can inform researchers who attempt to use Mamba to solve problems outside $\mathsf{TC}^0$. By identifying the constraints of the current design, our work pushed the exploration of the expressiveness of neural network models.

**Roadmap.** Section 2 introduces the works related to our paper. Section 3 introduces key computational concepts and Mamba definitions that form the basis for subsequent sections. Then, we present the circuit complexity results for Selective SSM and Mamba in Section 4. Section 5 details our hardness results. Finally, Section 6 gives a conclusion.

## 2. Related Work

**Complexity and Neural Network.** Circuit Complexity, a crucial set of metrics in computational complexity theory, studies the computational power of circuit families. It has valuable applications in comprehending the capabilities of machine learning models [21–32]. The complexity classes include $\mathsf{AC}^0$ represents problems that are highly parallelizable equipped with standard logic gates, which can be solved by constant-depth circuits with unbounded fan-in AND, OR, and NOT gates; $\mathsf{TC}^0$ class extends from $\mathsf{AC}^0$ with additional majority gates; $\mathsf{NC}^1$ problems can be solved by $O(\log n)$-depth circuits with bounded fan-in. These circuit complexity classes form a hierarchy: $\mathsf{AC}^0 \subset \mathsf{TC}^0 \subseteq \mathsf{NC}^1$ [24]. The question of whether $\mathsf{TC}^0 \neq \mathsf{NC}^1$ remains an open topic of discussion. [33] demonstrates that while Transformers can simulate nonsolvable semi-automata, their depth is influenced by the length of the input sequence. Building on this, [27] investigates the expressive power of Transformers augmented with Chain-of-Thought (CoT) reasoning in the context of circuit complexity. They propose the following relationships:

- $\mathsf{T}[\mathrm{poly}(n), 1, 1]$ is the subset of $\mathsf{CoT}[\log n, \mathrm{poly}(n), 1, 1]$ which is a subset of $\mathsf{AC}^0$.

- $\mathsf{T}[\mathrm{poly}(n), \log n, 1]$ is the subset of $\mathsf{CoT}[\log n, \mathrm{poly}(n), \log n, 0]$ which is a subset of $\mathsf{TC}^0$.

Here, $\mathsf{T}[d(n), s(n), e(n)]$ refers to a constant-depth Transformer with an embedding size of $d(n)$, precision $s(n)$ bits, and exponent size $e(n)$ for input length $n$. Meanwhile, $\mathsf{CoT}[T(n), d(n), s(n), e(n)]$ denotes a $T(n)$-step Chain-of-Thought process using a constant-depth Transformer $\mathsf{T}[d(n), s(n), e(n)]$. They use their framework to show that Transformers equipped with CoT are capable of tackling more complex problems. Therefore, circuit complexity has shown its effectiveness in representing the computational capabilities of neural networks.

**Limits on Transformers Model.** Transformers have shown outstanding performance on tasks from natural language processing, but they present limited effectiveness in mathematical computations. A series of research highlights the reasoning limitations of Transformer Model [20, 25, 34–38]. [20] shows that average-hard attention transformers (AHATs) and softmax-attention transformers (SMATs) are in DLOGTIME-uniform $\mathsf{TC}^0$ with $O(\mathrm{poly}(n))$-bit float number precision, indicating that they are equivalent to constant-depth threshold circuits with polynomial size, and their ability is limited when handling more complex reasoning tasks which require higher-depth or nonuniform computations. As a result, Transformers with SMATs or AHATs are inherently unable to solve problems outside $\mathsf{TC}^0$, especially those that involve many inherently sequential computations. What about Transformers with CoT? Even though Transformers with CoT can address relatively more problems than CoT, Transformers still fail to solve problems requiring reasoning beyond $\mathsf{TC}^0$.

**Architecture of State-Space Models (SSM).** SSMs have emerged as an alternative model to the popular LLMs, such as RNNs and Transformers. SSM presents ideal performance in tasks involving long-term dependencies and sequential reasoning [16]. The foundation of SSMs uses linear dynamical systems (LDS) or discrete-time state-space equations [16, 17] to represent the system's internal state and its evolution over time. Using these mechanisms, SSMs are able to capture the sequential nature of data by updating the state iteratively, which has efficient inference and state-tracking [39, 40]. Compared to RNNs, SSMs have better scalability and stability when handling long sequences, and SSMs are capable of resolving the gradient-related issues inherent to RNNs [16] and have recently garnered attention for their versatility across various tasks such as sequential recommendation [41, 42] and image deblurring [43].

Mamba is a recent advancement in SSM architecture, and it combines the efficient parallelizable computation from Transformers. SSMs in Mamba use kernel methods and spectral techniques to enable convolution and facilitate parallelizable computation [16, 17]. Mamba incorporates efficient memory embedding and long-term state representation into its architecture, making itself a strong opponent to the popular LLMs today, such as Transformers. However, despite the theoretical expectations of SSM and Mamba, it is crucial for us to understand the computational limits to conclude whether its capabilities outperform Transformers.

# 3. Preliminaries

In Section 3.1, we introduce the circuit complexity classes. In Section 3.2, we introduce the float point number. In Section 3.3, we introduce the Mamba block.

**Notation.** For $n \in \mathbb{Z}_+$, we define $[n] := \{1, 2, \ldots, n\}$. We use $\Pr[\cdot]$ to denote the probability. We use $\mathbb{E}[\cdot]$ to denote the expectation. We use $\mathrm{Var}[\cdot]$ to denote the variance. We define $\mathbf{1}_n \in \mathbb{R}^n$ as $(\mathbf{1}_n)_i := 1$, for all $i \in [n]$. Let $X_{i,j} \in \mathbb{R}$ be the $(i,j)$-th entry of an arbitrary matrix $X$. Let $\|X\|_\infty \in \mathbb{R}$ be the largest entry of the matrix $X$. We denote $x_i = \{0,1\}^*$ to be the binary sequence, where its length is not determined.

## 3.1. Circuit Complexity

In this section, we provide an introduction to the fundamental concepts of circuit complexity classes. We define the Boolean circuit below:

**Definition 3.1** (Boolean circuit, from Definition 6.1, On page 102 in [44]). *Let $n \in \mathbb{Z}_+$. A Boolean circuit with $n$ variables is represented on a directed acyclic graph and defined as a function $C_n : \{0,1\}^n \to \{0,1\}$. The graph's nodes represent logic gates, where input nodes (with in-degree 0) correspond to the $n$ Boolean variables. Each non-input gate computes its value based on the outputs provided by other connected gates.*

**Definition 3.2** (Circuit family recognizes languages, from Definition 6.2, On page 103 in [44]). *Let $x$ be an arbitrary element in $\{0,1\}^*$. Let $L$ be a subset of $\{0,1\}^*$ called a language.*

*If there is $C_{|x|} \in \mathcal{C}$ (a Boolean circuit) satisfying $C_{|x|}(x) = 1$ iff $x \in L$, then we say $L$ is recognized by a family $\mathcal{C}$ of Boolean circuits.*

We now introduce $\mathsf{NC}^i$ class.

**Definition 3.3** ($\mathsf{NC}^i$ [44]). $\mathsf{NC}^i$ *consists of languages that can be decided by Boolean circuits with a size of $O(\mathrm{poly}(n))$, depth $O((\log n)^i)$, and utilizing* OR, AND, *and* NOT *gates with bounded fan-in.*

When Boolean circuits are allowed to use AND and OR gates with unbounded fan-in, they become capable of recognizing a broader class of languages. The $\mathsf{AC}^i$ class is defined as follows.

**Definition 3.4** ($\mathsf{AC}^i$ [44]). $\mathsf{AC}^i$ *refers to the set of languages that Boolean circuits can recognize with size $O(\mathrm{poly}(n))$, depth $O((\log n)^i)$, and utilizing* AND, OR, *and* NOT *gates with unbounded fan-in.*

Since these three gates may be simulated by MAJORITY gates, we arrive at a broader complexity class, $\mathsf{TC}^i$.

**Definition 3.5** ($\mathsf{TC}^i$ [45]). $\mathsf{TC}^i$ *includes languages that can be recognized by Boolean circuits with size $O(\mathrm{poly}(n))$, depth $O((\log n)^i)$, and unbounded fan-in gates for* OR, AND, NOT, *and* MAJORITY. *A* MAJORITY *gate outputs 1 if more than half of its inputs are 1.*

**Remark 3.6.** *In Definition 3.5,* THRESHOLD *gates or* MOD *gates configured for prime values can replace* MAJORITY *gates. A Boolean circuit that includes any of these gates is referred to as a threshold circuit.*

**Definition 3.7** (P [44]). *A deterministic Turing machine in polynomial time with respect to the size of the input can recognize the languages in class* P.

**Fact 3.8** (Hierarchy Folklore, [44], From Corollary 4.35, On page 110 in [44], in [45]). *For all $i \in \mathbb{N}$, $\mathsf{NC}^i \subseteq \mathsf{AC}^i \subseteq \mathsf{TC}^i \subseteq \mathsf{NC}^{i+1} \subseteq \mathsf{P}$.*

**Remark 3.9.** *For $i = 0$, it is established that $\mathsf{NC}^0 \subsetneq \mathsf{AC}^0 \subsetneq \mathsf{TC}^0$. However, determining whether $\mathsf{TC}^0 \subsetneq \mathsf{NC}^1$ remains an open question in circuit complexity. Additionally, the question of whether $\mathsf{NC} := \cup_{i \in \mathbb{N}} \mathsf{NC}^i \subsetneq \mathsf{P}$ is also unresolved. For further discussion, see [44, 45].*

**Definition 3.10** (L-uniformity [44]). $\mathsf{C}$ *represents a language recognized by a circuit family $\mathcal{C}$, where $\mathcal{C}$ could be $\mathsf{NC}^i$, $\mathsf{AC}^i$, or $\mathsf{TC}^i$. Suppose we have a Turing machine that is satisfying for any arbitrary $n \in \mathbb{N}$, computes a circuit in $\mathcal{C}$ for $n$ variables from the input $1^n$ using $O(\log n)$ space, such that the circuit $C_n$ recognizes $L$, then a language $L$, which is the subset of $\{0,1\}^*$, is said to be in $\mathsf{L}$-uniform $\mathsf{C}$.*

We define DLOGTIME-uniformity and discuss the relationships between this definition and L-uniformity as follows.

**Definition 3.11** (DLOGTIME-uniformity in [46]). C *is defined as in Definition 3.10. Suppose we have a Turing machine that satisfying for any arbitrary $n \in \mathbb{N}$, computes $C_n$ in $\mathcal{C}$ for $n$ variables from the input $1^n$ within time $O(\log n)$, where $C_n$ recognizes $L$, then a language $L$, which is the subset of $\{0, 1\}^*$, is said to be in* DLOGTIME-*uniform* C.

## 3.2. Float Point Numbers

To compute SSM and Mamba correctly and effectively, we establish the computational framework by providing the definitions of the basic concepts of floating-point numbers and their related operations.

Notably, the operations provided below are not limited to purely theoretical work; in fact, they can be effectively realized in hardware.

**Lemma 3.12** (Efficient floating-point operations in $\mathsf{TC}^0$, Lemma 10, 11 in [20]). *Let $p \in \mathbb{Z}_+$. We have*

1. *We can use the uniform threshold circuit, which has the size of $\mathrm{poly}(n)$ and has a constant depth, to compute all $+, \cdot$, and comparison of two $p$-bit floating-point numbers, as defined in Definition A.3.*

2. *Using the same depth uniform threshold circuit as above, we can compute the iterative multiplication of $m$ numbers of floating-point numbers with $q$ bits.*

3. *Using the same depth uniform threshold circuit as above, we can compute the iterative addition of $m$ numbers of floating-point numbers with $q$ bits.*

*We use $d_{\mathrm{std}}, d_{\otimes}$, and $d_{\oplus}$ to denote the constant depth of the above three situations, respectively.*

**Corollary 3.13** (Floor operation in $\mathsf{TC}^0$). *Consider $p \in \mathbb{Z}_+$ being less than or equal to $\mathrm{poly}(n)$. We can implement the floor operation for a floating-point number with $q$ bits using the uniform threshold circuit, which has the size of $\mathrm{poly}(n)$ and has a constant depth $d_{\mathrm{std}}$.*

**Lemma 3.14** (Approximation of $\exp$ in $\mathsf{TC}^0$, Lemma 12 in [20]). *For any positive integer $p$ such that $p \leq \mathrm{poly}(n)$, there exists a uniform threshold circuit with size $\mathrm{poly}(n)$ and constant-depth that approximates $\exp(x)$ for any $p$-bit floating-point number $x$, with a relative error not exceeding $2^{-p}$. The depth required for this computation is denoted as $d_{\exp}$.*

**Lemma 3.15** (Approximation of square root in $\mathsf{TC}^0$, Lemma 12 in [20]). *Let $p$ be a positive integer satisfying $p \leq \mathrm{poly}(n)$. For any $p$-bit floating-point number $x$, a uniform threshold circuit with size $\mathrm{poly}(n)$ and constant-depth can compute $\sqrt{x}$ with a relative error of at most $2^{-p}$. The depth required for this computation is denoted as $d_{\mathrm{sqrt}}$.*

**Lemma 3.16** (Matrix multiplication, Lemma 4.2 in [20]). *Consider two matrices $A \in \mathbb{F}_p^{n_1 \times d}$ and $B \in \mathbb{F}_p^{d \times n_2}$. If $p, n_1, n_2, d \leq \mathrm{poly}(n)$, then we can use the uniform threshold circuit, which has the size of $\mathrm{poly}(n)$ and has a constant depth $(d_{\mathrm{std}} + d_{\oplus})$, to compute the product of $A$ and $B$.*

## 3.3. Mamba Blocks

Having established the necessary mathematical foundation, this section introduces the main components of the Mamba architecture, as illustrated in Figure 1. We start by discussing the input projection within the Mamba framework.

**Definition 3.17** (Mamba Input Projection). *Let $X \in \mathbb{F}_p^{L \times D}$ denote the input sequence, where $L$ is the sequence length, and $D$ is the feature dimension. We define the Mamba input projection function $\mathcal{L} : \mathbb{F}_p^{L \times D} \to \mathbb{F}_p^{L \times D'}$ as: $\mathcal{L}(X) := X \cdot W_x + \mathbf{1}_L b_x^\top$, where $W_x \in \mathbb{F}_p^{D \times D'}$ is the learned weight matrix, $b_x \in \mathbb{F}_p^{D'}$ is a learned bias vector, and $\mathbf{1}_L \in \mathbb{F}_p^{L \times 1}$ broadcasts $b_x$ across all rows.*

After the input projection, Mamba used a 1-D convolution layer to capture local temporal patterns by convolving the input features with a learned kernel.

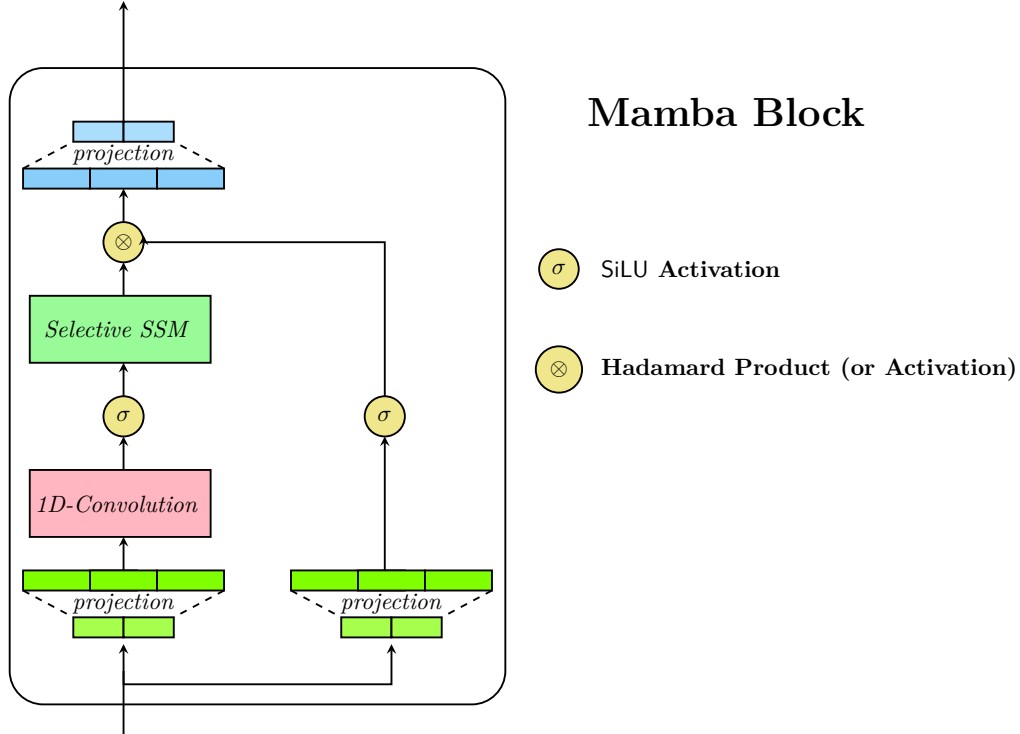

Figure 1: Mamba Block Architecture. The input is first processed through two input projections. One branch flows through an input projection, followed by a 1-D convolution, a SiLU activation, and a Selective SSM block before reaching the Hadamard product (or activation). The other branch passes through an input projection directly to a SiLU activation and then converges at the same Hadamard product (or activation). Finally, the output of the Hadamard product is passed through the output projection.

**Definition 3.18** (1-D Convolution). *Let $X \in \mathbb{F}_p^{L \times D'}$ denote the output of Definition 3.17, where $L$ is the sequence length and $D'$ is the projected feature dimension. Let $W \in \mathbb{F}_p^{K \times D' \times N}$ denote a convolutional kernel of size $K$, where $N$ is the number of output channels. We define the 1-D convolution layer function $\mathcal{C} : \mathbb{F}_p^{L \times D'} \to \mathbb{F}_p^{L \times N}$ as:*

$$\mathcal{C}(X)_{t,n} := \sum_{k=0}^{K-1} \sum_{d'=1}^{D'} W[k, d', n] \cdot X_{t-k,d'},$$

*for $t \in [L]$ and $n \in [N]$, where $X_{t-k,d'} = 0$ if $t - k < 0$, and zero-padding is applied for boundary cases; $W[k, d', n]$ selects the contribution of the $d'$-th feature at time step $t - k$ to the $n$-th output channel.*

Then, the convoluted input goes through a non-linear SiLU activation function in Mamba.

**Definition 3.19** (SiLU Activation). *Let $X \in \mathbb{F}_p^{L \times D} \cup \mathbb{F}_p^{L \times N}$ be the output from Definition 3.17 or Definition 3.18, where $B$ is the batch size, $L$ is the sequence length, and $D$ is the feature dimension. We define the entry wise SiLU function $\mathcal{Z} : \mathbb{F}_p^{L \times D} \cup \mathbb{F}_p^{L \times N} \to \mathbb{F}_p^{L \times D} \cup \mathbb{F}_p^{L \times N}$ as $\mathcal{Z}(X)_{t,d} := X_{t,d} \cdot \sigma(X_{t,d})$, where the sigmoid function $\sigma(X_{t,d}) : \mathbb{F}_p \to \mathbb{F}_p$ is defined as: $\sigma(X_{t,d}) := \frac{1}{1+e^{-X_{t,d}}}$. Here, $t \in [L]$ and $d \in [D]$ index the sequence and feature dimensions.*

Now, we introduce the softplus activation used in Mamba selection mechanisms as $\tau_\Delta$.

**Definition 3.20** (Softplus Activation). *We define* Softplus $: \mathbb{F}_p \to \mathbb{F}_p$ *as* Softplus$(z) := \log(1 + e^z)$.

Following this, the selection functions dynamically adapt the state-space parameters based on the input sequence, refining the model's ability to represent sequential dependencies by modulating the state-space matrices $B$, $C$, and $\Delta$ based on learned projection.

**Definition 3.21** (Selection Functions). *Let $X \in \mathbb{F}_p^{L \times D}$ denote the input sequence. Let $\tau_\Delta = \mathsf{Softplus}(w_\Delta)$, where $w_\Delta \in \mathbb{F}_p$ is a learned scalar, and $\mathsf{Softplus}$ is given in Definition 3.20. The selection functions $s_B : \mathbb{F}_p^{L \times D} \to \mathbb{F}_p^{n \times N}$, $s_C : \mathbb{F}_p^{L \times D} \to \mathbb{F}_p^{D' \times N}$, $s_\Delta : \mathbb{F}_p^{L \times D} \to \mathbb{F}_p$ are defined as:*

$$s_B(X) := W^B X P^B, \quad s_C(X) := W^C X P^C, \quad and\ s_\Delta(X) := \tau_\Delta \cdot \mathsf{Broadcast}_D(W^\Delta X P^\Delta),$$

*where $W^B \in \mathbb{F}_p^{n \times L}$, $W^C \in \mathbb{F}_p^{D' \times L}$, and $W^\Delta \in \mathbb{F}_p^{1 \times L}$ are learned selection weight matrices, $P^B \in \mathbb{F}_p^{D \times N}$, $P^C \in \mathbb{F}_p^{D \times N}$, $P^\Delta \in \mathbb{F}_p^D$ are projection matrices, and the function $\mathsf{Broadcast}_D : \mathbb{F}_p \to \mathbb{F}_p$ replicates the result of $W^\Delta X P^\Delta$ across all feature dimensions.*

With the selection functions implemented, we now introduce the Selective SSM in Mamba.

**Definition 3.22** (Selective SSM in Mamba). *Let $X \in \mathbb{F}_p^{L \times N}$ be the output of Definition 3.18. Given a diagonal matrix $A \in \mathbb{F}_p^{n \times n}$, we define the Selective SSM function $\mathsf{SSM}_{\mathrm{select}} : \mathbb{F}_p^{L \times N} \to \mathbb{F}_p^{L \times D'}$ as $\mathsf{SSM}_{\mathrm{select}}(X) := \mathsf{SSM}_{\mathrm{recur}}(X, A, s_B(X), s_C(X), s_\Delta(X))$, where $\mathsf{SSM}_{\mathrm{recur}}(X) \in \mathbb{F}_p^{L \times D'}$ is the recurrent SSM output from Definition A.6, and $s_B(X), s_C(X), s_\Delta(X)$ are selection mechanisms from Definition 3.21.*

Finally, we introduce the Mamba output projection, which maps the processed sequence back to the original feature dimension.

**Definition 3.23** (Mamba Output Projection). *Let $X \in \mathbb{F}_p^{L \times D'}$ denote the output from Definition 3.22, where $L$ is the sequence length and $D'$ is the feature dimension. We define the Mamba output projection function $\mathcal{O} : \mathbb{F}_p^{L \times D'} \to \mathbb{F}_p^{L \times D}$ as:*

$$\mathcal{O}(X) := X \cdot W_x + \mathbf{1}_L b_x^\top,$$

*where $W_x \in \mathbb{F}_p^{D' \times D}$ is the learned weight matrix, $b_x \in \mathbb{F}_p^D$ is a learned bias vector, and $\mathbf{1}_L \in \mathbb{F}_p^{L \times 1}$ broadcasts $b_x$ across all rows.*

Through this progression, we can now define Mamba as a series of composite functions.

**Definition 3.24** (Mamba). *Let $X \in \mathbb{F}_p^{L \times D}$ denote the input sequence, where $L$ is the sequence length, and $D$ is the feature dimension. We define the Mamba architecture function $\mathcal{M} : \mathbb{F}_p^{L \times D} \to \mathbb{F}_p^{L \times D}$ as:*

$$\mathcal{M}(X) = \mathcal{O}((\mathsf{SSM}_{\mathrm{select}} \circ \mathcal{Z} \circ \mathcal{C} \circ \mathcal{L}(X)) \otimes (\mathcal{Z} \circ \mathcal{L}(X)),$$

*where $\circ$ is function composition, $\mathcal{L}$ is Mamba Input Projection (see Definition 3.17), $\mathcal{C}$ is 1-D Convolution Layer (see Definition 3.18), $\mathcal{Z}$ is $\mathsf{SiLU}$ Activation (see Definition 3.19), $\mathsf{SSM}_{\mathrm{select}}$ is Selective SSM (see Definition 3.22), $\otimes$ is Hadamard Product or Activation, and $\mathcal{O}$ is Mamba Output Projection (see Definition 3.23).*

# 4. Complexity of SSM and Mamba

In Section 4.1, we provide an approximation of the logarithm function within $\mathsf{TC}^0$. In Section 4.2, we analyze the complexity of computing Recurrent SSM. In Section 4.3, we investigate the complexity of computing Convolution SSM. In Section 4.4, we establish circuit complexity bounds for selective SSM. In Section 4.5, we present the circuit complexity bounds for Mamba computations.

## 4.1. Approximating Logarithm in $\mathsf{TC}^0$

In this section, we show the approximation of logarithm can be done in $\mathsf{TC}^0$ circuit. The logarithm function is a key component of the Softplus activation function, which plays a central role in the selection mechanisms of the Selective SSM within the Mamba architecture. Therefore, the ability to compute logarithm in $\mathsf{TC}^0$ is crucial for ensuring Selective SSM and Mamba operate within constant depth $\mathsf{TC}^0$.

**Lemma 4.1** (Approximating Logarithm in $\mathsf{TC}^0$, informal version of Lemma B.3). *For any $p$-bit floating-point number $x \in \mathbb{F}_p$, we can use a uniform threshold circuit, where the depth is $d_{\log}$ and the size is $\mathrm{poly}(n)$, the logarithm $\log(x)$, where the relative error is less than or equal to $2^{-p}$.*

*Sketch of the proof.* To approximate $\log(x)$, we normalize $x = \langle m, e \rangle$ into $r \in [\frac{1}{2}, 1]$ or $r \in [1, 2]$, depending on whether $e$ is even or odd. This normalization adjusts the exponent to $k$ and can be computed by $\mathsf{TC}^0$ circuit in constant depth.

We use Taylor series expansion around 1 to approximate $\log(r)$, and we can get an approximation of $\log(r)$ with relative error bounded by $2^{-p-1}$. Using the same technique, we can approximate $\log(2)$. Lastly, we compute $\log(x)$ as $\log(x) = \log(r) + k \cdot \log(2)$. The $\mathsf{TC}^0$ circuit in constant depth can compute all operations. □

## 4.2. Recurrent SSMs are in $\mathsf{TC}^0$

In this section, we show recurrent SSM is in $\mathsf{TC}^0$. We provide more details about recurrent SSM in Appendix A.2.

**Lemma 4.2** (Recurrent SSM in $\mathsf{TC}^0$). *Let $C \in \mathbb{F}_p^{D' \times n}$, $\mathcal{H}(X, A, B, \Delta) \in \mathbb{F}_p^{L \times n}$, and $X \in \mathbb{F}_p^{L \times N}$ denote the input matrix and intermediate computations, where $p, L, N, n, D' \leq \mathrm{poly}(n)$. We can use a uniform threshold circuit, where the depth is $d_{\mathrm{recur}}$ and the size is $\mathrm{poly}(n)$, to compute the Recurrent SSM function $\mathsf{SSM}_{\mathrm{recur}}(X, A, B, C, \Delta) \in \mathbb{F}_p^{L \times D'}$, as defined in Definition A.6.*

*Proof.* From Definition A.6, the Recurrent SSM computation is given by:

$$\mathsf{SSM}_{\mathrm{recur}}(X, A, B, C, \Delta)_{t,d} := \sum_{i=1}^{n} \overline{C}_{d,i} \cdot \mathcal{H}(X, A, B, \Delta)_{t,i},$$

The computation of $\mathsf{SSM}_{\mathrm{recur}}(X)$ involves two primary steps: computing the hidden state updates $\mathcal{H}(X, A, B, \Delta)$ and iterative addition with multiplication. We use a threshold circuit whose depth is

- $d_h$ to compute $\mathcal{H}(X, A, B, \Delta)$ (Lemma B.6),
- $d_{\mathrm{std}}$ to compute $\overline{C}_{d,i} \cdot \mathcal{H}(X, A, B, \Delta)_{t,i}$ (Lemma 3.12),
- $d_{\oplus}$ to compute $\sum_{i=1}^{n} \overline{C}_{d,i} \cdot \mathcal{H}(X, A, B, \Delta)_{t,i}$ (Lemma 3.12)

Finally, we can show: $d_{\mathrm{recur}} = d_h + (d_{\mathrm{std}} + d_{\oplus})$. Therefore, we get our desired result. □

## 4.3. Convolution SSMs are in $\mathsf{TC}^0$

In this section, we show convolution SSM is in $\mathsf{TC}^0$. We provide more details about recurrent SSM in Appendix A.3.

**Lemma 4.3** (Convolution SSM in $\mathsf{TC}^0$). *Let $\overline{K} \in \mathbb{F}_p^{D' \times D \times M}$, $X \in \mathbb{F}_p^{L \times N}$, where $p, L, N, D', M \leq \mathrm{poly}(n)$. We can use a threshold circuit, where the depth is $d_{\mathrm{conv}}$ and the size is $\mathrm{poly}(n)$, to compute the convolution SSM $\mathsf{SSM}_{\mathrm{conv}} : \mathbb{F}_p^{L \times N} \times \mathbb{F}_p^{n \times n} \times \mathbb{F}_p^{n \times D} \times \mathbb{F}_p^{D' \times n} \times \mathbb{F}_p \to \mathbb{F}_p^{L \times D'}$, as defined in Definition A.8.*

*Proof.* From Definition A.8, the convolution output sequence is given by:

$$\mathsf{SSM}_{t,d}^{\mathrm{conv}}(X, A, B, C, \Delta) = \sum_{k=0}^{L-1} \sum_{d=1}^{D} \overline{K}[d', d, k] \cdot X_{t-k,d}.$$

It can be computed as follows. Using a threshold circuit, we can perform

- matrix multiplication to compute $\sum_{d=1}^{D} \overline{K}[d', d, k] \cdot X_{t-k,d}$ (Lemma 3.16) and

- iterated addition to compute $\sum_{k=0}^{L-1} \sum_{d=1}^{D} \overline{K}[d', d, k] \cdot X_{t-k,d}$ (Lemma 3.12),

whose depths are $d_{\text{std}} + d_{\oplus}$ and $d_{\oplus}$, respectively. Finally, we can conclude that: $d_{\text{conv}} = d_{\text{std}} + 2d_{\oplus}$. Thus, we get the desired result. $\qquad\square$

## 4.4. Circuit Complexity Bound for Selective SSM

In this section, we formulate the circuit complexity bound for Selective SSM.

**Theorem 4.4** (Selective SSM in $\mathsf{TC}^0$)**.** *Let $X \in \mathbb{F}_p^{L \times N}$ represent the output sequence from* SiLU *activated 1-D convolution layer* (*see Definition 3.18*), *where $L$ is the sequence length and $N$ is the number of output channels, with $L, N \leq \text{poly}(n)$. We may use a uniform threshold circuit, whose depth is $d_{\text{SSM}}$ and size is* $\text{poly}(n)$, *to compute the Selective SSM* (*Definition 3.22*).

*Proof.* The Selective SSM combines the selection functions, discretization, and state-space dynamics, which we have already proved to be in $\mathsf{TC}^0$.

To compute Selective SSM, we can follow the following. Using a threshold circuit, we can compute

- selection functions (Lemma B.10),
- discretization (Lemma B.2)
- recurrent SSM (Lemma 4.2), or
- convolution SSM (Lemma 4.3)

whose depths are $d_{\text{select}}$, $d_{\text{disc}}$, $d_{\text{recur}}$, and $d_{\text{conv}}$ respectively. Finally, we can show:

$$d_{\text{SSM}} = d_{\text{select}} + d_{\text{disc}} + d_{\text{recur}} \text{ for recurrent SSM,}$$
$$d_{\text{SSM}} = d_{\text{select}} + d_{\text{disc}} + d_{\text{conv}} \text{ for convolution SSM.}$$

Therefore, we get our desired result. $\qquad\square$

## 4.5. Circuit Complexity Bound for Mamba

In this section, we formulate the circuit complexity bound for Mamba.

**Theorem 4.5** (Main property for Mamba)**.** *Let $X \in \mathbb{F}_p^{L \times D}$ represent the input sequence, where $L$ is the sequence length and $D$ is the feature dimension, with $L, D \leq \text{poly}(n)$. We may use a uniform threshold circuit, whose depth is $d_{\text{mamba}}$ and size is $\text{poly}(n)$, to compute the Mamba architecture.*

*Proof.* The Mamba from Definition 3.24 is given:

$$\mathcal{M}(X) = \mathcal{O}((\mathsf{SSM}_{\text{select}} \circ \mathcal{Z} \circ \mathcal{C} \circ \mathcal{L}(X)) \otimes (\mathcal{Z} \circ \mathcal{L}(X)),$$

Using a threshold circuit, we can compute

- input projections (Lemma 3.16) using matrix multiplication and addition,
- 1-D Convolution (Lemma B.9),
- entrywise SiLU (Lemma B.5),
- Selective SSM (Theorem 4.4),
- Hadamard Product (Lemma B.1),
- output projection (Lemma 3.16) using matrix multiplications and additions,

whose depths are $d_{\text{std}} + d_{\oplus}$, $d_{\text{1dconv}}$, $d_{\exp} + d_{\text{std}}$, $d_{\text{select}}$, $d_{\text{std}}$, and $d_{\text{std}} + d_{\oplus}$, respectively.

Finally, we can show $d_{\text{mamba}} = d_{\text{1dconv}} + d_{\exp} + d_{\text{select}} + 4d_{\text{std}} + d_{\oplus}$

Therefore, we can get the desired result. □

Theorem 4.5 demonstrates that a DLOGTIME-uniform $\mathsf{TC}^0$ circuit family can simulate Mamba, showing the Mamba representation capacity limitations. In previous work, [19] showed that SSM and Mamba can be simulated by L-uniform $\mathsf{TC}^0$ with $c \log(n)$ precision. However, we improve the uniformity and precision in [19] by proving that Mamba can be simulated by DLOGTIME-uniform $\mathsf{TC}^0$ with $\text{poly}(n)$ precision by new techniques introduced from [20]. Our complexity bound is better than previous work.

## 5. Hardness

In this section, we present the hardness result: Selective SSM and Mamba, which are constrained in $\mathsf{TC}^0$, cannot solve problems residing in $\mathsf{NC}^1$, such as arithmetic formula evaluation, Boolean formula value problems, and permutation composition. These results show the limitations of Selective SSM and Mamba in their expressive power.

**Theorem 5.1** (Informal proof of Theorem C.22). *if $\mathsf{TC}^0 \neq \mathsf{NC}^1$, float point number is $\text{poly}(n)$-bits precision, layers are constant-depth, and hidden dimension is $O(n)$ size, then we can have the Selective SSM and Mamba are not capable of resolving the arithmetic formula evaluation problems, boolean formula value problem, and permutation composition problems.*

*Proof Sketch.* To show Selective SSM and Mamba cannot solve arithmetic formula evaluation problems, Boolean formula value problems, and permutation composition problems. We leverage the difference between the complexity classes $\mathsf{TC}^0$ and $\mathsf{NC}^1$, under the assumption $\mathsf{TC}^0 \neq \mathsf{NC}^1$. Arithmetic formula evaluation problems, Boolean formula value problems, and permutation composition problems are defined to be $\mathsf{NC}^1$ problems in Section C.1, C.2, and C.3. From previous proof, we show Selective SSM and Mamba are both in $\mathsf{TC}^0$. Therefore, they cannot solve those $\mathsf{NC}^1$ problems. □

To the best of our knowledge, there is no previous work proving that Mamba and SSM with $\text{poly}(n)$ precision cannot solve arithmetic formula problems, boolean formula value problems, and permutation composition problems.

## 6. Conclusion

In this paper, we conducted a rigorous mathematical analysis of the computational limits of SSM and Mamba. We use the framework of circuit complexity and demonstrate that Mamba and SSMs, despite their stateful designs, fall into DLOGTIME-uniform $\mathsf{TC}^0$ with $\text{poly}(n)$-precision. These results show that SSM and Mamba are fundamentally equivalent to Transformers in terms of computational expressiveness, as their architectures are all constrained by the complexity class $\mathsf{TC}^0$. As a result, Mamba cannot solve problems outside $\mathsf{TC}^0$, such as arithmetic formula evaluation and Boolean formula value problems, unless $\mathsf{TC}^0 = \mathsf{NC}^1$.

Our contributions include formal proofs of the circuit complexity bounds for Mamba and SSMs, and we show that their computational performances are equivalent to constant-depth uniform threshold circuits. Additionally, we provide hardness results. The hardness results show that these architectures cannot resolve sequential and state-dependent tasks that require higher computational depth. These new findings challenge the assumption that Mamba has higher computational capabilities than Transformers. By building the theoretical limits of Mamba and SSMs, our work contributes to the broader understanding of the computational power of modern neural network models. We emphasize the need for future innovations to solve problems beyond $\mathsf{TC}^0$ so they can solve more complex and inherently sequential problems. We hope our study can inspire more research on designing newer architectures that can balance efficiency, scalability, and enhanced expressiveness.

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

# Appendix

**Roadmap.** In Section A, we introduce more definitions related to our work, including circuit complexity definitions, float point operations, and definitions for recurrent and convolution SSM. In Section B, we present more proofs of the components of our main Theorem 4.4 and 4.5. In Section C, we present the definitions for our hardness problems and the results with Selective SSM and Mamba. In Section D, we provide more related works.

## A. Preliminaries

In this section, we introduce more definitions related to our work. In Section A.1, we introduce more float point numbers and their operations. In Section A.2, we define the components of Recurrent SSM. In Section A.3, we define the components of Convolution SSM.

We begin by introducing the notations used in this paper.

**Notation** For $n \in \mathbb{Z}_+$, we define $[n] := \{1, 2, \ldots, n\}$. We use $\Pr[\cdot]$ to denote the probability. We use $\mathbb{E}[\cdot]$ to denote the expectation. We use $\mathrm{Var}[\cdot]$ to denote the variance.

We define $\mathbf{1}_n \in \mathbb{R}^n$ as $(\mathbf{1}_n)_i := 1$, for all $i \in [n]$. Let $X_{i,j} \in \mathbb{R}$ be the $(i,j)$-th entry of an arbitrary matrix $X$. Let $\|X\|_\infty \in \mathbb{R}$ be the largest entry of the matrix $X$. We denote $x_i = \{0, 1\}^*$ to be the binary sequence, where its length is not determined.

### A.1. Float Point Numbers

In this section, we introduce the float point numbers.

**Definition A.1** (Floating-point number, From Definition 9 in [20]). *A $p$-bit floating-point number is defined as a pair $\langle m, e \rangle$, where $m$ (the significand) is an integer satisfying $m \in (-2^p, -2^{p-1}) \cup \{0\} \cup [2^{p-1}, 2^p)$, and $e$ (the exponent) is an integer within the range $e \in [-2^p, 2^p)$. The value of the floating-point number $\langle m, e \rangle$ corresponds to the real number $m \cdot 2^e$. The set of all $p$-bit floating-point numbers is denoted as $\mathbb{F}_p$.*

**Definition A.2** (Rounding, From Definition 9 in [20]). *$x$ is a floating point or in $\mathbb{R}$. Let $\mathrm{round}_p(x)$ be a floating-point number with $p$-bit closest to $x$ with an even significand in case of a tie.*

**Definition A.3** (Floating-point number operations, [20]). *Consider $a, b \in \mathbb{Z}$. Let the operation $a \mathbin{/\!\!/} b$ be as follows. Suppose $a/b = C1/4$, where $C \in \mathbb{Z}$, then $a \mathbin{/\!\!/} b = a/b$. Or, $a \mathbin{/\!\!/} b$ is equal to $a/b + 1/8$.*

*With floating points $\langle m_1, e_1 \rangle$, $\langle m_2, e_2 \rangle$ having $p$-bits, we define the following operations:*

- *addition:*

$$\langle m_1, e_1 \rangle + \langle m_2, e_2 \rangle := \begin{cases} \mathrm{round}_p(\langle m_1 + m_2 \mathbin{/\!\!/} 2^{e_1 - e_2}, e_1 \rangle) & \text{if } e_1 \geq e_2, \\ \mathrm{round}_p(\langle m_1 \mathbin{/\!\!/} 2^{e_2 - e_1} + m_2, e_2 \rangle) & \text{if } e_1 \leq e_2, \end{cases}$$

- *multiplication:*

$$\langle m_1, e_1 \rangle \times \langle m_2, e_2 \rangle := \mathrm{round}_p(\langle m_1 m_2, e_1 + e_2 \rangle)$$

- *division:*

$$\langle m_1, e_1 \rangle \div \langle m_2, e_2 \rangle := \mathrm{round}_p(\langle m_1 2^{p-1} \mathbin{/\!\!/} m_2, e_1 - e_2 - p + 1 \rangle)$$

- *comparison:*

$$\langle m_1, e_1 \rangle \leq \langle m_2, e_2 \rangle \leftrightarrow \begin{cases} m_1 \leq m_2 \mathbin{/\!\!/} 2^{e_1 - e_2} & \text{if } e_1 \geq e_2, \\ m_1 \mathbin{/\!\!/} 2^{e_2 - e_1} \leq m_2 & \text{if } e_1 \leq e_2. \end{cases}$$

- *floor: if $e \geq 0$, then $\lfloor \langle m, e \rangle \rfloor := \langle m 2^e, 0 \rangle$. If $e < 0$, then $\lfloor \langle m, e \rangle \rfloor := \mathrm{round}(\langle m/2^{-e}, 0 \rangle)$*

## A.2. Discretization: Recurrent SSM

In this section, we define and formalize the discretization of recurrent SSMs and their associated components. We provide a structured foundation for understanding their functionality and computation. We begin by introducing the discrete transformation technique that transforms the continuous state-space representations into discrete ones.

**Definition A.4** (Discrete State Space Transformation). *Let $\Delta$ denote the discretization step size. The discrete parameters $\overline{A} \in \mathbb{F}_p^{n \times n}$, $\overline{B} \in \mathbb{F}_p^{n \times D}$, and $\overline{C} \in \mathbb{F}_p^{D' \times n}$ are defined as follows:*

$$\overline{A} := \exp(\Delta A),$$
$$\overline{B} := (\Delta A)^{-1}(\exp(\Delta A) - I) \cdot \Delta B,$$
$$\overline{C} := C,$$

*where $\exp(\Delta A)$ denotes the matrix exponential of $\Delta A$, $A \in \mathbb{F}_p^{n \times n}$ is the continuous state transition matrix, $B \in \mathbb{F}_p^{n \times D}$ is the continuous input influence matrix, $C \in \mathbb{F}_p^{D' \times n}$ is the output projection matrix, and $I \in \mathbb{F}_p^{n \times n}$ is the identity matrix.*

Transitioning from the discretization step, we proceed to the hidden state recurrence in recurrent SSM, which is the core update mechanism for hidden states across timesteps.

**Definition A.5** (Hidden State Recurrence). *Let $H \in \mathbb{F}_p^{L \times n}$ denote the hidden state, and $X \in \mathbb{F}_p^{L \times N}$ be the output of Definition 3.18, where $L$ is the length of the sequence and $n$ denotes the hidden state dimensions. We define the hidden state update function $\mathcal{H} : \mathbb{F}_p^{L \times N} \times \mathbb{F}_p^{n \times n} \times \mathbb{F}_p^{n \times D} \times \mathbb{F}_p \to \mathbb{F}_p^{L \times n}$ as:*

$$\mathcal{H}(X, A, B, \Delta)_{t,i} := \sum_{j=1}^{n} \overline{A}_{i,j} \cdot H_{t-1,j} + \sum_{k=1}^{D} \overline{B}_{i,k} \cdot X_{t,k},$$

*where $\overline{A} \in \mathbb{F}_p^{n \times n}$ and $\overline{B} \in \mathbb{F}_p^{n \times D}$ are the parameters from Definition A.4, $H_{t-1,j}$ denotes the hidden state at timestep $t-1$, initialized as $H_{0,i} = 0$, and $X_{t,k}$ denotes the input matrix at timestep t.*

Finally, we are able to formalize recurrent SSMs, which combine the hidden state update mechanism with the output projection step.

**Definition A.6** (Recurrent SSM). *Let $X \in \mathbb{F}_p^{L \times N}$ be the output of Definition 3.18. We define the Recurrent SSM function $\mathrm{SSM}_{\mathrm{recur}} : \mathbb{F}_p^{L \times N} \times \mathbb{F}_p^{n \times n} \times \mathbb{F}_p^{n \times D} \times \mathbb{F}_p^{D' \times n} \times \mathbb{F}_p \to \mathbb{F}_p^{L \times D'}$ as:*

$$\mathrm{SSM}_{\mathrm{recur}}(X, A, B, C, \Delta)_{t,d} := \sum_{i=1}^{n} \overline{C}_{d,i} \cdot \mathcal{H}(X, A, B, \Delta)_{t,i},$$

*where $\mathcal{H}(X) \in \mathbb{F}_p^{L \times n}$ is the hidden state update function defined in Definition A.5, and $\overline{C} \in \mathbb{F}_p^{D' \times n}$ is the output projection matrix, mapping the hidden state to the output space.*

## A.3. Discretization: Convolutional SSM

In this section, we extend the formulation of SSM by presenting its convolutional implementations after discretization. These are the core mechanisms that enable its parallel computations. We first show the kernel computation.

**Definition A.7** (Convolution Kernel). *Let $\overline{A} \in \mathbb{F}_p^{n \times n}$, $\overline{B} \in \mathbb{F}_p^{n \times D}$, and $\overline{C} \in \mathbb{F}_p^{D' \times n}$ denote the discrete state-space parameters. We define the convolution kernel $\overline{K} \in \mathbb{F}_p^{D' \times D \times M}$ for parallel computations as:*

$$\overline{K}[d', d, k] = \sum_{i=1}^{n} \sum_{j=1}^{n} \overline{C}_{d',i} \cdot (\overline{A}^k)_{i,j} \cdot \overline{B}_{j,n},$$

*where $d' \in [D']$ is the output feature dimension index, $d \in [D]$ is the input feature dimension index, and $k \in [M]$ is the time offset index, and $M$ is the length of the kernel.*

By using this kernel $\overline{K}$, we can compute the final output sequence through convolution.

**Definition A.8** (Convolution Output Sequence for SSM). *Let $X \in \mathbb{F}_p^{L \times N}$ be the output from Definition 3.18), where $t \in [L]$ is the index of the sequence, $d \in [D]$ is the index of input feature. Using the kernel $\overline{K} \in \mathbb{F}_p^{D' \times D \times M}$ from Definition A.7, we define the convolution SSM $\mathrm{SSM}_{\mathrm{conv}} : \mathbb{F}_p^{L \times N} \times \mathbb{F}_p^{n \times n} \times \mathbb{F}_p^{n \times D} \times \mathbb{F}_p^{D' \times n} \times \mathbb{F}_p \to \mathbb{F}_p^{L \times D'}$ as:*

$$\mathrm{SSM}_{t,d}^{\mathrm{conv}}(X, A, B, C, \Delta) = \sum_{k=0}^{L-1} \sum_{d=1}^{D} \overline{K}[d', d, k] \cdot X_{t-k,d}$$

*for each $t = 0, 1, \ldots, L-1$, Here $\mathrm{SSM}_{t,d}^{\mathrm{conv}}$ is the output for timestep $t$ and output feature $d$, $\overline{K}[d', d, k]$ is the kernel weight for output feature $d'$, input feature $d$, and time offset $k$, and $X_{t-k,d}$ is the input for timestep $t-k$, and input dimension $d$.*

# B. Complexity of SSM and Mamba

In this section, we provide additional proofs to support our theorem.

In Section B.1, we show the Hadamard product is in $\mathsf{TC}^0$. In Section B.2, we show the discretization in SSM is in $\mathsf{TC}^0$. In Section B.3, we show approximating logarithm can be done in $\mathsf{TC}^0$. In Section B.4, we show the Softplus Activation is in $\mathsf{TC}^0$. In Section B.5, we show the SiLU Activation is in $\mathsf{TC}^0$. In Section B.6, we show the hidden state update function is in $\mathsf{TC}^0$. In Section B.7, we show the computation of kernel in Convolution SSM is in $\mathsf{TC}^0$. In Section B.8, we show the convolution indexing is in $\mathsf{TC}^0$. In Section B.9, we show the 1-D convolution layer in Mamba is in $\mathsf{TC}^0$. In Section B.10, we show the selective functions are in $\mathsf{TC}^0$.

## B.1. Computing Entry-wise Matrix Multiplication

Now, we present computing entrywise matrix multiplication.

**Lemma B.1** (Hadamard Product in $\mathsf{TC}^0$). *Let $A \in \mathbb{F}_p^{n \times d}$ and $B \in \mathbb{F}_p^{n \times d}$. If $p \leq \mathrm{poly}(n)$, $n \leq \mathrm{poly}(n)$, and $d \leq n$, then we can compute the Hadamard product $A \circ B$ using a uniform threshold circuit, whose depth is $d_{\mathrm{std}}$, and size is $\mathrm{poly}(n)$.*

*Proof.* We have $(A \circ B)_{i,j} = A_{i,j} \cdot B_{i,j}$. By Lemma 3.12, a threshold circuit with constant depth $d_{\mathrm{std}}$ can compute every product $A_{i,j} \cdot B_{i,j}$. Since the computations of $A_{i,j} \cdot B_{i,j}$ for different pairs $(i, j)$ are independent, all such products can be computed in parallel with the same depth $d_{\mathrm{std}}$.

The circuit's size stays polynomial in $n$ because both $n$ and $d$ are bounded by $\mathrm{poly}(n)$, and each multiplication is implemented using a circuit of poly size. $\square$

## B.2. Computing Discretization

In this section, we prove computing discretization is in $\mathsf{TC}^0$.

**Lemma B.2** (Discretization in $\mathsf{TC}^0$). *Let $A \in \mathbb{F}_p^{n \times n}$ be a diagonal matrix and $B \in \mathbb{F}_p^{n \times d}$, where $n \leq \mathrm{poly}(n)$, and $d \leq \mathrm{poly}(n)$. Then a uniform threshold circuit with size $\mathrm{poly}(n)$ and constant depth $d_{\mathrm{disc}}$ can compute the discrete parameters $\overline{A}$ and $\overline{B}$ from Definition A.4.*

*Proof.* Given the discretization parameter:

$$\overline{A} := \exp(\Delta A),$$
$$\overline{B} := (\Delta A)^{-1}(\exp(\Delta A) - I) \cdot \Delta B.$$

The computation involves three main steps: computing $\exp(\Delta A)$, inverting $\Delta A$, and performing matrix multiplications.

Since $A$ is diagonal, each entry of $\exp(\Delta A)$ can be computed independently as $(\exp(\Delta A))_{i,i} = \exp(\Delta A_{i,i})$. By part 1 of Lemma 3.12 and Lemma 3.14, $\overline{A}$ can be computed in depth-$(d_{\mathrm{std}} + d_{\exp})$.

To compute $(\Delta A)^{-1}$, each entry of $(\Delta A)^{-1}$ can be computed independently as $((\Delta A)^{-1})_{i,i} = (\Delta A_{i,i})^{-1}$. By part 1 of Lemma 3.12, this inversion is in depth-$d_{\mathrm{std}}$.

Next, we compute $\overline{B}$ as follows: To compute $\exp(\Delta A) - I$, each entry $(\exp(\Delta A) - I)_{i,i} = \exp(\Delta A_{i,i}) - 1$ can be computed independently in depth-$d_{\exp} + d_{\mathrm{std}}$ by Lemma 3.12 and Lemma 3.14; to compute $(\Delta A)^{-1} \cdot (\exp(\Delta A) - I)$, since both matrices are diagonal, we perform element-wise multiplication, which uses depth-$d_{\mathrm{std}}$ by Lemma B.1; to compute $(\Delta A)^{-1} \cdot (\exp(\Delta A) - I) \cdot B$, we perform matrix multiplication, which uses depth-$d_{\mathrm{std}} + d_{\oplus}$.

Finally, we can show

$$d_{\mathrm{disc}} = 5d_{\mathrm{std}} + 2d_{\exp} + d_{\oplus}$$

The circuit's size stays polynomial in $n$ because both $n$ and $d$ are bounded by $\mathrm{poly}(n)$, and each operation is implemented using a circuit of poly size. $\qquad\square$

## B.3. Approximating Logarithm in $\mathsf{TC}^0$

In this Section, we present the formal proof for approximating logarithm in $\mathsf{TC}^0$

**Lemma B.3** (Approximate Logarithm in $\mathsf{TC}^0$, formal version of Lemma 4.1). *For any $p$-bit floating-point number $x \in \mathbb{F}_p$, we can use a uniform threshold circuit, whose depth is $d_{\log}$ and size is $\mathrm{poly}(n)$ to approximate the logarithm $\log(x)$, where the error is less than or equal to $2^{-p}$.*

*Proof.* We can use truncated Taylor Series ([47, 48]).

Let $p \in O(\mathrm{poly}(n))$. For $\log(x)$ where $x = \langle m, e \rangle$: If $e$ is even, let $r = m \cdot 2^{-p} \in [\frac{1}{2}, 1)$ and $k = e + p$; otherwise, let $r = m \cdot 2^{-p+1} \in [1, 2)$ and $k = e + p - 1$.

Compute $\log(r)$ using the Taylor series about 1:

$$\log(r) = \sum_{i=1}^{N-1} (-1)^{i+1} \frac{(r-1)^i}{i} + O(|r-1|^N).$$

Since $|r - 1| < 1$, there is an $N \in O(p)$ that makes the relative error at most $2^{-p-1}$. Then we compute $\log(x)$ as follows:

$$\log(x) = \log(r) + k \cdot \log(2).$$

To compute $\log(2)$, use the Taylor series:

$$\log 2 = \sum_{i=1}^{N-1} \frac{1}{i \cdot 2^i} + O(2^{-N}).$$

Thus, we approximate $\log(x)$ as:

$$\log(x) \approx \sum_{i=1}^{N-1} (-1)^{i+1} \frac{(r-1)^i}{i} + k \cdot \sum_{i=1}^{N-1} \frac{1}{i \cdot 2^i}.$$

Since $N \in O(p)$, the total error is less than or equal to $2^{-p}$.

We can determine the total depth of the circuit required for these computations using Lemma 3.12. To normalize $x$ and compute the value of $k$, we must perform the division and floor operations, both of which can be executed using a circuit of depth $d_{\mathrm{std}}$; to compute $\log(r)$ using Taylor series, we perform iterated multiplication, addition, and iterated addition, which uses a depth-$d_{\oplus} + d_{\otimes} + d_{\mathrm{std}}$ circuit; to compute $k \cdot \log(2)$, we perform iterated multiplication, addition, and iterated addition, which uses a depth-$d_{\oplus} + d_{\otimes} + d_{\mathrm{std}}$ circuit; to compute $\log(x)$, we perform addition, which uses a depth-$d_{\mathrm{std}}$

Finally, we can show

$$d_{\log} = 2d_{\oplus} + 2d_{\otimes} + 3d_{\mathrm{std}}.$$

Thus, we complete the proof. □

## B.4. Computing the Softplus Activation

In this section, we show the proof for Computing the Softplus Activation is in $\mathsf{TC}^0$

**Lemma B.4** (Softplus in $\mathsf{TC}^0$). *For any $x \in \mathbb{F}_p$, size $\mathrm{poly}(n)$ and constant depth $d_{\mathrm{sp}}$ uniform threshold circuit, we can approximate the Softplus function, as defined in Definition 3.20, where the error is less than or equal to $2^{-p}$.*

*Proof.* $\mathsf{Softplus}(z) = \log(1 + e^z)$ can be calculated as the following. To compute $\exp(z)$, we perform exponential function, which uses a depth-$d_{\exp}$ by Lemma 3.14; to compute $1 + \exp(z)$, we perform addition, which uses a depth-$d_{\mathrm{std}}$ by Part 1 from Lemma 3.12; to compute $\log(1 + \exp(z))$, we perform logarithm, which uses a depth-$d_{\log}$ by Lemma B.3

Finally, we can show

$$d_{\mathrm{sp}} = d_{\exp} + d_{\mathrm{std}} + d_{\log}.$$

Therefore, using the uniform threshold circuit, where its size is equal to $\mathrm{poly}(n)$ and its depth is $d_{\mathrm{sp}}$, we can compute $\mathsf{Softplus}(z)$. □

## B.5. Computing the SiLU Activation

In this section, we show the proof of SiLU, used in Mamba is in $\mathsf{TC}^0$.

**Lemma B.5** (SiLU Activation in $\mathsf{TC}^0$). *Let $z \in F_p^D$ denote the input feature vector, where $p, D \leq \mathrm{poly}(n)$. The SiLU defined in Definition 3.19 is computed using a uniform threshold circuit, where its size is equal to $\mathrm{poly}(n)$ and its depth is $(d_{\exp} + d_{\mathrm{std}})$.*

*Proof.* From Definition 3.19, SiLU is given as

$$\mathsf{SiLU} = z \cdot \sigma(z),$$

where $\sigma(z)$ denotes the sigmoid function, defined as:

$$\sigma(z) = \frac{1}{1 + e^{-z}}.$$

We compute $\mathsf{SiLU}(z)$ as follows. To compute $e^{-z}$, we use Lemma 3.14, and it can be computed by a threshold circuit in depth-$d_{\exp}$; to compute $z \cdot \frac{1}{1 + e^{-z}}$, we perform addition, division, and multiplication. By Part 1 from Lemma 3.12, we can compute it using a threshold circuit in depth-$d_{\mathrm{std}}$.

Therefore, we get the desired result. □

## B.6. Hidden State Recurrent in $\mathsf{TC}^0$

In this section, we prove the hidden state update in Recurrent SSM is in $\mathsf{TC}^0$.

**Lemma B.6** (Hidden State Recurrence in $\mathsf{TC}^0$). *Let $A \in \mathbb{F}_p^{n \times n}$, $B \in \mathbb{F}_p^{n \times D}$, and $X \in \mathbb{F}_p^{L \times D}$ denote the input matrix, where $p, n, D \leq \mathrm{poly}(n)$. The hidden state recurrence from Definition A.5 can be computed by a threshold circuit with size $\mathrm{poly}(n)$ and constant depth $d_h$.*

*Proof.* From Definition A.5, the hidden state recurrence is given by:

$$\mathcal{H}(X, A, B, \Delta)_{t,i} := \sum_{j=1}^{n} \overline{A}_{i,j} \cdot H_{t-1,j} + \sum_{k=1}^{D} \overline{B}_{i,k} \cdot X_{t,k},$$

where $\overline{A} \in \mathbb{F}_p^{n \times n}$, $\overline{B} \in \mathbb{F}_p^{n \times D}$, $H \in \mathbb{F}_p^{L \times n}$ is the hidden state, and $X \in \mathbb{F}_p^{L \times D}$ is the input sequence.

The computation of $\mathcal{H}(X, A, B, \Delta)$ involves two steps: iterative addition, multiplication, and addition:

To compute $\sum_{j=1}^{n} \overline{A}_{i,j} \cdot H_{t-1,j}$ and $\sum_{k=1}^{D} \overline{B}_{i,k} \cdot X_{t,k}$, we need multiplication and iterated addition. By Lemma 3.12, we can compute them by a threshold circuit in depth-$d_{\text{std}} + d_\oplus$; to compute $\sum_{j=1}^{n} \overline{A}_{i,j} \cdot H_{t-1,j} + \sum_{k=1}^{D} \overline{B}_{i,k} \cdot X_{t,k}$, we then perform addition. By Lemma 3.12, it can be computed by a threshold circuit in depth-$d_{\text{std}}$

The total depth of the circuit for computing $\mathcal{H}(X, A, B, \Delta)$ is given by:

$$d_h = 2d_{\text{std}} + d_\oplus.$$

Since the circuit size is polynomial in $n$ and the depth $d_h$ is constant, we get our desired result. $\square$

## B.7. Computing Kernel in Convolution SSMs is in $\mathsf{TC}^0$

In this section, we show the computation of Kernel in $\mathsf{TC}^0$.

**Lemma B.7** (Convolution Kernel in $\mathsf{TC}^0$). *Let $\overline{A} \in \mathbb{F}_p^{n \times n}$, $\overline{B} \in \mathbb{F}_p^{n \times D}$, and $\overline{C} \in \mathbb{F}_p^{D' \times n}$, where $p, n, D, D', M \leq \text{poly}(n)$. The convolution kernel $\overline{K} \in \mathbb{F}_p^{D' \times D \times M}$, as defined in Definition A.7, can be computed by a threshold circuit with size $\text{poly}(n)$ and constant depth $d_k$.*

*Proof.* From Definition A.7, the convolution kernel computation is given by:

$$\overline{K}[d', d, k] = \sum_{i=1}^{n} \sum_{j=1}^{n} \overline{C}_{d',i} \cdot (\overline{A}^k)_{i,j} \cdot \overline{B}_{j,n},$$

We can compute in the following steps

1. Since $\overline{A}$ is a diagonal matrix, each entry $(\overline{A}^k)_{i,i}$ can be computed as $(\overline{A}_{i,i})^k$. By part 2 of Lemma 3.12, iterated multiplication can be computed by a threshold circuit with constant depth $d_\otimes$. The computations of $(\overline{A}_{i,i})^k$ for all $i$ are independent, so $\overline{A}^k$ can be computed in depth $d_\otimes$.

2. To compute $(\overline{A}^k \cdot \overline{B})$, we perform matrix multiplication. By Lemma 3.16, we can compute it using a threshold circuit where its depth is $d_{\text{std}} + d_\oplus$.

3. To compute $\overline{K}[d', d, k]$, it performs another matrix multiplication $\overline{C} \cdot (\overline{A}^k \cdot \overline{B})$. By Lemma 3.16, we can compute it using a threshold circuit where its depth is $d_{\text{std}} + d_\oplus$.

Finally, we can show that

$$d_k = d_\otimes + 2d_{\text{std}} + 2d_\oplus,$$

so we get the desired result. $\square$

## B.8. Convolution Indexing in $\mathsf{TC}^0$

In this section, we prove the indexing operation in 1-D Convolution is in $\mathsf{TC}^0$.

**Lemma B.8** (Convolution Indexing in $\mathsf{TC}^0$). *Let $X \in \mathbb{F}_p^{L \times D}$ denote the input sequence, where $L$ is the sequence length, and $D$ is the feature dimension. Let $t \in [L]$ and $k \in [K]$ denote indices for time steps and kernel offsets. $L, D, K \leq \text{poly}(n)$. Retrieving the value $X_{t-k,d}$ for $b \in [B]$ and $d \in [D]$, with zero-padding applied for $t - k < 0$, can be computed by a uniform threshold circuit with size $\text{poly}(n)$ and constant depth $d_{\text{std}}$.*

*Proof.* The indexing operation has two primary operations: checking the boundary and retrieving the value.

To compute boundary checking for each time step $t \in [L]$, kernel offset $k \in [K]$, and feature $d \in [D]$, we need to check if $t - k < 0$ for the zero-padding. We define $\mathsf{BoundaryCheck}(t, k)$ function as follows:

$$\mathsf{BoundaryCheck}(t, k) = \begin{cases} 1 & \text{if } t - k < 0, \\ 0 & \text{otherwise.} \end{cases}$$

To compute $\mathsf{BoundaryCheck}(t, k)$, we perform subtraction and comparison. By Part 1 from lemma 3.12, they can be computed in $d_{\mathrm{std}}$.

To compute value retrieval, we can establish the following:

$$X_{t-k,d} = (1 - \mathsf{BoundaryCheck}(t, k)) \cdot X_{t-k,d}$$

where if $\mathsf{BoundaryCheck}(t, k) = 1$, $X_{t-k,d}$ will be evaluated to $0$ so we apply zero padding.

To compute $X_{t-k,d}$, we perform subtraction and multiplication. By Part 1 from Lemma 3.12, they can be computed in $d_{\mathrm{std}}$.

Therefore, we get the desired result. $\qquad\square$

## B.9. 1-D Convolution in $\mathsf{TC}^0$

In this section, we show the 1-D convolution layer in Mamba is in $\mathsf{TC}^0$.

**Lemma B.9** (1-D Convolution in $\mathsf{TC}^0$)**.** *Let $W \in \mathbb{F}_p^{K \times D' \times N}$ and $X \in \mathbb{F}_p^{L \times D'}$, where $p, K, L, D', N \leq \mathrm{poly}(n)$. We can use the threshold circuit, where its size is $\mathrm{poly}(n)$ and its depth is $d_{\mathrm{1dconv}}$ to compute the 1-D convolution function $\mathcal{C} : \mathbb{F}_p^{L \times D'} \to \mathbb{F}_p^{L \times N}$ (see Definition 3.18).*

*Proof.* The 1-d convolution from Definition 3.18 is the following:

$$\mathcal{C}(X)_{t,n} = \sum_{k=0}^{K-1} \sum_{d'=1}^{D'} W[k, d', n] \cdot X_{t-k,d'},$$

this convolution has three primary operations: matrix indexing, entry-wise multiplications, and summation.

We can compute $\mathcal{C}(X)$ as the following. To compute matrix indexing, from Lemma B.8, it can be computed with a threshold circuit in depth-$d_{\mathrm{std}}$; to compute $\sum_{d'=1}^{D'} W[k, d', n] \cdot X_{t-k,d'}$ for kernel index $k \in [K]$ and feature dimension $d' \in [D']$, we perform matrix multiplication. By Lemma 3.16, it can be computed with a threshold circuit with depth-$d_{\mathrm{std}} + d_\oplus$; to compute $\sum_{k=0}^{K-1} \sum_{d'=1}^{D'} W[k, d', n] \cdot X_{t-k,d'}$, we perform iterated addition. By Part 1 from Lemma 3.12, it can be computed with a threshold in depth-$d_\oplus$.

Finally, we can show that

$$d_{\mathrm{1dconv}} = 2d_{\mathrm{std}} + 2d_\oplus.$$

Therefore, we get the desired result. $\qquad\square$

## B.10. Selection Functions in $\mathsf{TC}^0$

In this section, we show selective functions computation are in $\mathsf{TC}^0$.

**Lemma B.10** (Selection Functions in $\mathsf{TC}^0$)**.** *Let $X \in \mathbb{F}_p^{L \times D}$ denote the input sequence. Let $W^B \in \mathbb{F}_p^{n \times L}$, $W^C \in \mathbb{F}_p^{D' \times L}$, and $W^\Delta \in \mathbb{F}_p^{1 \times L}$ denote learned selection weight matrices, and $P^B \in \mathbb{F}_p^{D \times N}$, $P^C \in \mathbb{F}_p^{D \times N}$, $P^\Delta \in \mathbb{F}_p^{D}$ denote projection matrices. We can use the threshold circuit, where its size is $\mathrm{poly}(n)$ and its depth is $d_{\mathrm{select}}$ to compute the selection function (see Definition 3.21).*

*Proof.* The selection mechanisms from Definition 3.21 are the following $s_B(X) = W^B X P^B, s_C(X) = W^C X P^C, s_\Delta(X) = \tau_\Delta \cdot \mathsf{Broadcast}_D(W^\Delta X P^\Delta),.$

These computations have three main operations: matrix multiplications, broadcasting, and non-linear activations.

We can compute selection functions as follows. To compute both $s_B(X) = W^B X P^B, s_C(X) = W^C X P^C$, and $W^\Delta X P^\Delta$, we perform matrix multiplication. By Lemma 3.16, we compute it using the threshold circuit (where the depth is $d_{\mathrm{std}} + d_\oplus$); to compute $\mathsf{Broadcast}(W^\Delta X P^\Delta)$, we simply copying the scalar value across $D$ dimensions, which is a simple duplication operation in constant depth-$d_{\mathrm{dup}}$; to compute $\tau_\Delta$ which is $\mathsf{Softplus}(w_\Delta)$ in this case, by Lemma B.4, it can be computed by a threshold circuit in depth-$d_{\mathrm{sp}}$; to compute $\tau_\Delta \cdot \mathsf{Broadcast}_D(W^\Delta X P^\Delta)$, we perform multiplication. By Part 1 from Lemma 3.12, it can be computed by a threshold circuit in depth-$d_{\mathrm{std}}$.

Finally, we can show

$$d_{\mathrm{select}} = 2d_{\mathrm{std}} + d_\oplus + d_{\mathrm{dup}} + d_{\mathrm{sp}}.$$

Therefore, we get our desired result. $\square$

# C. Our Hardness Results

We present the problems about the arithmetic formula in Section C.1. We analyze the Boolean formula value problem in Section C.2. We introduce the permutation composition problem in Section C.3. In Section C.4, we state our four hardness results.

## C.1. The First Problem

Now, we show the following definition from [49].

**Definition C.1** (Arithmetic formula, Definition in [49]). *Let $\mathbb{S}$ be a semi-ring (which may also be a ring or field). An arithmetic formula over $\mathbb{S}$ with indeterminates $X_1, X_2, \ldots, X_n$ is defined by:*

- *For $i \in [n]$, $X_i$ is an arithmetic formula.*

- *For every $c \in \mathbb{S}$, $c$ is an arithmetic formula.*

- *If $\alpha$ is an arithmetic formula and $\theta$ is a unary operation of $\mathbb{S}$ then $(\theta\alpha)$ is arithmetic formula.*

- *If $\alpha$ and $\beta$ are arithmetic formulas and $\theta$ is a binary operator of $\mathbb{S}$ then $(\alpha\theta\beta)$ is an arithmetic formula.*

*An arithmetic formula $A$ with indeterminates $X_1, \ldots, X_n$ is denoted by $A(X_1, \ldots, X_n)$.*

After defining the arithmetic formula, we then present its computational implications.

**Definition C.2** (Arithmetic formula evaluation problem, Definition in [49]). *Let $\mathbb{S}$ be a ring, field, or semi-ring. The arithmetic formula evaluation problem is: Given an arithmetic formula $A(X_1, X_2, \ldots, X_n)$ over $\mathbb{S}$ and constants $c_1, c_2, \ldots, c_n \in \mathbb{S}$, what is $A(c_1, c_2, \ldots, c_n)$?*

**Remark C.3.** *In [49], they have shown that the problem defined in Definition C.2 belongs to $\mathsf{NC}^1$.*

## C.2. The Second Problem

In this section, we show the second problem.

**Definition C.4** (Definition in [50], page 1). *We have $\Sigma = \{0, 1, \wedge, \vee, \neg, (,)\}$. We define the Boolean formula by the following:*

- *We have $0$ and $1$ being the Boolean formulas.*

- *Suppose we have $\beta, \alpha$ being the Boolean formulas. Then, we can get that $(\alpha \wedge \beta)$, $(\neg\alpha)$, and $(\alpha \vee \beta)$ being the Boolean formulas.*

Also, we define the following

**Definition C.5** (Definition in [50]. page 1). *We define $|\alpha|$ to be the amount of symbols from $\alpha$ (which is a string).*

**Definition C.6** (Definition in [50]. page 1). *We define the Boolean formula by the following:*

- *We have $0$ and $1$ being the Boolean formulas.*

- *Suppose we have $\beta$ being the Boolean formulas. Then, we can get that $(\alpha\neg)$ being the Boolean formulas.*

- *Suppose we have $\beta, \alpha$ being the Boolean formulas. Suppose $|\alpha|$ is greater than or equal to $|\beta|$. Then, we can get that $\alpha\beta\wedge$ and $\alpha\beta\vee$ are the Boolean formulas.*

We use $0$ to denote False and $1$ to denote True.

**Lemma C.7** (Page 1 in [50]). *Consider a problem that decides the Boolean formula's true value. This problem falls in $\mathsf{NC}^1$.*

## C.3. Permutation Composition Problem

In this section, we present the permutation composition problem as established in [51] and its computational implications.

**Definition C.8** (Permutation, based on [51]). *A permutation is a bijection $\pi : [n] \to [n]$, where $[n] = \{1, 2, \ldots, n\}$. The set of all permutations on $[n]$ forms a group $S_n$, called the symmetric group. A permutation $\pi \in S_n$ may be represented in standard forms such as cycle notation or pointwise mapping.*

**Definition C.9** (Permutation composition, based on [51]). *The composition of two permutations $\pi_1, \pi_2 \in S_n$ is the permutation $\pi = \pi_2 \circ \pi_1$, defined by $\pi(x) = \pi_2(\pi_1(x))$ for all $x \in [n]$. The composition of a sequence of permutations $\pi_1, \pi_2, \ldots, \pi_k \in S_n$ is given by:*

$$\Pi = \pi_k \circ \pi_{k-1} \circ \cdots \circ \pi_1.$$

**Definition C.10** (Permutation composition problem, based on [51]). *The permutation composition problem is defined as if there is a sequence of permutations $\pi_1, \pi_2, \ldots, \pi_k \in S_n$ represented in a standard form, then the result of the composition $Pi = \pi_k \circ \pi_{k-1} \circ \cdots \circ \pi_1$ is expressed in the same representation.*

**Definition C.11** (Word problem for permutations, based on [51]). *A specific instance of the permutation composition problem is the word problem for permutations. This problem is defined as if there is a sequence of permutations $\pi_1, \pi_2, \ldots, \pi_k \in S_n$, then we need to determine whether $\Pi = \pi_k \circ \pi_{k-1} \circ \cdots \circ \pi_1$ equals the identity permutation $e$, where $e(x) = x$ for all $x \in [n]$.*

The following theorems highlight the significance of the permutation composition problem within computational complexity:

**Lemma C.12** (Theorem 1 in [51]). *Any language recognized by a fan-in 2 Boolean circuit of depth $d = O(\log n)$ can be recognized by a width-5 permutation branching program (PBP) of polynomial size. Consequently, the class of languages recognized by polynomial-size PBPs of bounded width equals $\mathsf{NC}^1$.*

**Lemma C.13** (Word Problem Completeness, based on [51]). *The word problem for the group $S_5$, which involves determining whether a composition of permutations equals the identity, is $\mathsf{NC}^1$-complete under $\mathsf{AC}^0$ reductions.*

## C.4. Results About Hardness

We introduce the hardness results for arithmetic formula evaluation problems.

**Lemma C.14.** *if $\mathsf{TC}^0 \neq \mathsf{NC}^1$, float point number is $\mathrm{poly}(n)$-bits precision, layers are constant-depth, and hidden dimension is $O(n)$ size, then we can have that Definition C.2 cannot be solved by the SSM.*

*Proof.* It is by Theorem 4.4, Lemma C.3, and Fact 3.8. □

**Lemma C.15.** *if* $\mathsf{TC}^0 \neq \mathsf{NC}^1$, *float point number is* $\mathrm{poly}(n)$*-bits precision, layers are constant-depth, and hidden dimension is* $O(n)$ *size, then we can have that Definition C.2 cannot be solved by the Mamba.*

*Proof.* It is by Theorem 4.5, Lemma C.3, and Fact 3.8. $\qquad\square$

We introduce the hardness results for the Boolean formula problem.

**Lemma C.16.** *if* $\mathsf{TC}^0 \neq \mathsf{NC}^1$, *float point number is* $\mathrm{poly}(n)$*-bits precision, layers are constant-depth, and hidden dimension is* $O(n)$ *size, then we can have that Definition C.6 cannot be solved by the SSM.*

*Proof.* It is by Theorem 4.4, Lemma C.7, and Fact 3.8. $\qquad\square$

**Lemma C.17.** *if* $\mathsf{TC}^0 \neq \mathsf{NC}^1$, *float point number is* $\mathrm{poly}(n)$*-bits precision, layers are constant-depth, and hidden dimension is* $O(n)$ *size, then we can have that Definition C.6 cannot be solved by the Mamba.*

*Proof.* It is by Theorem 4.5, Lemma C.7, and Fact 3.8. $\qquad\square$

We introduce the hardness results for permutation composition problems.

Here, we show SSM and Mamba cannot solve Width-5 PBPs from Lemma C.12.

**Lemma C.18.** *If* $\mathsf{TC}^0 \neq \mathsf{NC}^1$, *float point number is* $\mathrm{poly}(n)$*-bits precision, layers are constant-depth, and hidden dimension is* $O(n)$ *size, then we can have the SSM cannot solve the Width-5 PBPs.*

*Proof.* It is by Theorem 4.4, Lemma C.12, and Fact 3.8. $\qquad\square$

**Lemma C.19.** *If* $\mathsf{TC}^0 \neq \mathsf{NC}^1$, *float point number is* $\mathrm{poly}(n)$*-bits precision, layers are constant-depth, and hidden dimension is* $O(n)$ *size, then we can have the Mamba cannot solve the Width-5 PBPs.*

*Proof.* It is by Theorem 4.5, Lemma C.12, and Fact 3.8. $\qquad\square$

Here, we show SSM and Mamba cannot solve the word problem from Lemma C.13.

**Lemma C.20.** *If* $\mathsf{TC}^0 \neq \mathsf{NC}^1$, *float point number is* $\mathrm{poly}(n)$*-bits precision, layers are constant-depth, and hidden dimension is* $O(n)$ *size, then we can have the SSM cannot solve the word problem.*

*Proof.* It is by Theorem 4.4, Lemma C.13, and Fact 3.8. $\qquad\square$

**Lemma C.21.** *If* $\mathsf{TC}^0 \neq \mathsf{NC}^1$, *float point number is* $\mathrm{poly}(n)$*-bits precision, layers are constant-depth, and hidden dimension is* $O(n)$ *size, then we can have the Mamba cannot solve the word problem.*

*Proof.* It is by Theorem 4.5, Lemma C.13, and Fact 3.8. $\qquad\square$

**Theorem C.22** (Formal proof of Theorem 5.1). *if* $\mathsf{TC}^0 \neq \mathsf{NC}^1$, *float point number is* $\mathrm{poly}(n)$*-bits precision, layers are constant-depth, and hidden dimension is* $O(n)$ *size, then we can have the Selective SSM and Mamba cannot solve the arithmetic formula evaluation problems, boolean formula value problem, and permutation composition problems.*

*Proof.* Based on Lemma C.14, C.15, C.16, C.17, C.18, C.19, C.20, and C.21.

We conclude the Selective SSM and Mamba cannot solve the Definition C.6 and Definition C.2, and permutation composition problems.

Thus, we complete the proof. $\qquad\square$

# D. More Related Work

**Theoretical Machine Learning.** Our work also takes inspiration from the following Machine Learning Theory work. Some works analyze the expressiveness of a neural network using the theory of complexity [52–58]. Some works optimize the algorithms that can accelerate the training of a neural network [58–84]. Some works analyze neural networks via regressions [85–95]. Some works use reinforcement learning to optimize the neural networks [96–102]. Some works optimize the attention mechanisms [103, 104].

**Accelerating Attention Mechanisms.** The attention mechanism, with its quadratic computational complexity concerning context length, encounters increasing challenges as sequence lengths grow in modern large language models [11, 105–108]. To address this limitation, polynomial kernel approximation methods [109] have been introduced, leveraging low-rank approximations to efficiently approximate the attention matrix. These methods significantly enhance computation speed, allowing a single attention layer to perform both training and inference with nearly linear time complexity [110, 111]. Moreover, these techniques can be extended to advanced attention mechanisms, such as tensor attention, while retaining almost linear time complexity for both training and inference [112]. [113] provides an almost linear time algorithm to accelerate the inference of VAR Transformer. Other innovations include RoPE-based attention mechanisms [114, 115] and differentially private cross-attention approaches [116]. Alternative strategies, such as the conv-basis method proposed in [104], present additional opportunities to accelerate attention computations, offering complementary solutions to this critical bottleneck. Additionally, various studies explore pruning-based methods to expedite attention mechanisms [117–124].

**Gradient Approximation.** The low-rank approximation is a widely utilized approach for optimizing transformer training by reducing computational complexity [111, 118, 125–128]. Building on the low-rank framework introduced in [110], which initially focused on forward attention computation, [111] extends this method to approximate attention gradients, effectively lowering the computational cost of gradient calculations. The study in [125] further expands this low-rank gradient approximation to multi-layer transformers, showing that backward computations in such architectures can achieve nearly linear time complexity. Additionally, [126] generalizes the approach of [111] to tensor-based attention models, utilizing forward computation results from [112] to enable efficient training of tensorized attention mechanisms. Lastly, [127] applies low-rank approximation techniques during the training of Diffusion Transformers (DiTs), demonstrating the adaptability of these methods across various transformer-based architectures.

