# OpenReview forum: "The Computational Limits of State-Space Models and Mamba via the Lens of Circuit Complexity"
_CPAL.cc/2025/Proceedings_Track — CPAL 2025 (Proceedings Track) Oral_

### Official Review · Reviewer_Gz7j · 2025-01-12
**Very insightful finding**

**Rating:** 7
**Confidence:** 1

**Review:**

This paper gives a through proof on the computation limit of state space model and Mamba. Comparing to traditional transformers, the state space models like Mamba are not more computationally expressive. The design of state space models is not fundamentally superior comparing to transformers and I think this is very insightful finding. This finding can encourage people to explore more network structures.

---

### Official Review · Reviewer_LVkZ · 2025-01-14

**Rating:** 7
**Confidence:** 2

**Review:**

**Summary**

This paper uses the circuit complexity framework to analyze the computational limits of Mamba and State-space Models (SSMs). The main result theoretically shows that Mamba has the computational capabilities as Transformers, refuting the popular belief that Mamba and SSMs may be strong candidates to outperform Transformers

**Quality/Clarity**
The paper is well written. In particular,
1) The introduction does a good job of presenting the problem the paper aims to address.
2) The preliminaries present the necessary background in sufficient detail to understand the subsequent results.

**Strengths**
- The paper does a very good job of breaking down and presenting all the fundamental/building block results necessary to understand the main result of the paper
- I believe this paper is a nice contribution to the field. It answers an important and refutes a popular held belief about the performance differences between Transformers and SSM and Mamba.

**Weaknesses**
- I am having a hard time identifying which results in section 4 and 5 are new results vs. those that are previously known. Could the authors clarify this for me.
- It seems that Theorem 5.1 is the main result of the paper, yet the authors spend very little time discussing the theorem, its proof sketch, and implications. I think removing some of the proof sketches of earlier results and spending more time/discussion in section 5 would greatly improve the paper.

**Minor comments/questions**
- Are some results of section 4 previous known results or new contributions? For example, approximate logarithm in TC^0.

---

### Meta-Review · Area_Chair_qTo8 · 2025-02-02

**Recommendation:** Accept (Poster)
**Confidence:** 3

**Metareview:**

The paper analyzes the computational limits of Mamba and State-Space Models (SSMs) using a circuit complexity framework, challenging the belief that these models surpass Transformers in computational expressiveness. The reviewers found the paper generally well-written and insightful, with a solid theoretical contribution, clear problem presentation, and a thorough breakdown of key concepts. However, some concerns were raised regarding the clarity of novel contributions and the discussion of the main results. I recommend acceptance and encourage the authors to address these concerns in the camera-ready version.

---

### Decision · Program_Chairs · 2025-02-11

Accept (Oral)